# TELEClass: Taxonomy Enrichment and LLM-Enhanced Hierarchical Text Classification with Minimal Supervision

## Abstract

Hierarchical text classification aims to categorize each document into a set of classes in a label taxonomy, which is a fundamental web text mining task with broad applications such as web content analysis and semantic indexing. Most earlier works focus on fully or semi-supervised methods that require a large amount of human annotated data which is costly and time-consuming to acquire. To alleviate human efforts, in this paper, we work on hierarchical text classification with a minimal amount of supervision: using the sole class name of each node as the only supervision. Recently, large language models (LLM) have shown competitive performance on various tasks through zero-shot prompting, but this method performs poorly in the hierarchical setting because it is ineffective to include the large and structured label space in a prompt. On the other hand, previous weakly-supervised hierarchical text classification methods only utilize the raw taxonomy skeleton and ignore the rich information hidden in the text corpus that can serve as additional class-indicative features. To tackle the above challenges, we propose TELEClass, Taxonomy Enrichment and LLM-Enhanced weakly-supervised hierarchical text Classification, which combines the general knowledge of LLMs and task-specific features mined from an unlabeled corpus. TELEClass automatically enriches the raw taxonomy with class-indicative features for better label space understanding and utilizes novel LLM-based data annotation and generation methods specifically tailored for the hierarchical setting. Experiments show that TELEClass can significantly outperform previous strong baselines while also achieving comparable performance to zero-shot prompting of LLMs with drastically less inference cost.

## CCS Concepts

• **Information systems** → **Data mining**; • **Computing methodologies** → **Natural language processing**; **Classification and regression trees**.

## Keywords

Weakly-Supervised Text Classification, Hierarchical Text Classification, Taxonomy Enrichment, Large Language Model

**ACM Reference Format:**
Anonymous Author(s). 2018. TELEClass: Taxonomy Enrichment and LLM-Enhanced Hierarchical Text Classification with Minimal Supervision. In

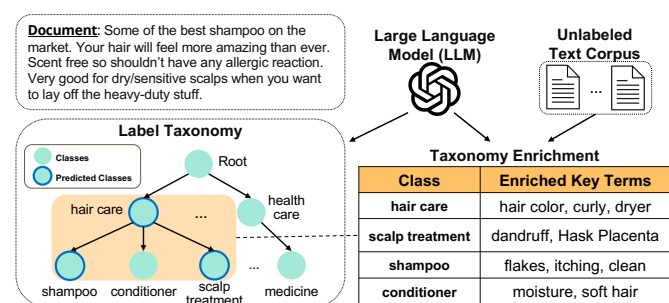

**Figure 1: An example document tagged with 3 classes. We automatically enrich each node with class-indicative terms and utilize LLMs to facilitate classification.**

*Proceedings of Make sure to enter the correct conference title from your rights confirmation emai (Conference acronym 'XX).* ACM, New York, NY, USA, 10 pages. https://doi.org/XXXXXXX.XXXXXXX

## 1 Introduction

Hierarchical text classification, aiming to classify documents into one or multiple classes in a label taxonomy, is a fundamental task in web text mining and NLP. Compared with standard text classification where label space is flat and relatively small (e.g., less than 20 classes), hierarchical text classification is more challenging given the larger and more structured label space and the existence of fine-grained and long-tail classes. Hierarchical text classification has broad applications such as web content organization [13], semantic indexing [25, 29], and query classification [7, 21, 23]. Recent studies also show that hierarchically structured text [14, 42] and document-level tagging [38] can improve retrieval-augmented generation for large language models.

The key challenge of hierarchical text classification is how to understand the large structured label space to distinguish the semantics of similar classes. Most earlier works tackle this task in fully supervised [9, 24, 56] or semi-supervised settings [5, 20], and different models are proposed to learn from a substantial amount of human-labeled data. However, acquiring human annotation is often costly, time-consuming, and not scalable.

Recently, large language models (LLM) such as GPT-4 [36] and Claude 3 [2] have demonstrated strong performance in flat text classification [48]. However, applying LLMs in hierarchical settings remains challenging [57]. Directly including hundreds of classes in prompts is ineffective and inefficient, leading to structural information loss, diminished clarity for LLMs at distinguishing class-specific information, and prohibitively expensive inference cost given the long prompt for each test document.

Along another line of research, Meng et al. [34] propose to train a moderate-size text classifier by utilizing a small set of keywords or

labeled documents for each class and a large unlabeled corpus. However, compiling keyword lists for hundreds of classes and obtaining representative documents for each specific and niche category still demand significant human efforts. Shen et al. [44] study the hierarchical text classification with *minimal supervision*, which takes the class name as the only supervision signal. Specifically, they introduce TaxoClass which generates pseudo labels with a textual entailment model for classifier training. However, this method overlooks additional class-relevant features in the corpus that could be helpful for label space understanding. It also suffers from the unreliable pseudo label selection because the entailment model is not trained to compare which class is more relevant to the document.

In this study, we advance minimally supervised hierarchical text classification by taking the advantage of both LLMs' text understanding ability and task-specific knowledge of the unlabeled text corpus. First, we tackle the challenge of label space understanding by enriching the label taxonomy with class-specific terms derived from two sources: LLM generation and automated extraction from the corpus. For example, the "conditioner" class in Figure 1 is enriched with key terms like "moisture" and "soft hair", which distinguish it from other classes. These terms enhance the supervision signal by combining knowledge from LLMs and text corpus and improve the pseudo label quality for classifier training. Second, we improve LLMs' ability in hierarchical text classification from two perspectives: we enhance LLM annotation efficiency and effectiveness through a taxonomy-guided candidate search and also optimize LLM-based document generation to create more precise pseudo data by using taxonomy paths.

Leveraging the above ideas, we introduce TELEClass: Taxonomy Enrichment and LLM-Enhanced weakly-supervised hierarchical text Classification. TELEClass consists of four major steps: (1) *LLM-Enhanced Core Class Annotation*, where we identify document "core classes" (i.e., fine-grained classes that most accurately describe the documents) by first enriching the taxonomy with LLM-generated key terms and then finding candidate classes with a top-down tree search algorithm for LLM to select the most precise core classes. (2) *Corpus-Based Taxonomy Enrichment*, where we analyze the taxonomy structure to additionally identify class-indicative topical terms through semantic and statistical analysis on the corpus. (3) *Core Class Refinement with Enriched Taxonomy*, where we embed documents and classes based on the enriched label taxonomy and refined the initially selected core classes by identifying the most similar classes for each document. (4) *Text Classifier Training with Path-Based Data Augmentation*, where we sample label paths from the taxonomy and guide the LLM to generate pseudo documents most accurately describing these fine-grained classes. Finally, we train the text classifier on two types of pseudo labels, the core classes and the generated data, with a simple text matching network and multi-label training strategy.

The contributions of this paper are summarized as follows:

- We propose TELEClass, a new method for minimally supervised hierarchical text classification, which requires only the class names of the label taxonomy as supervision to train a multi-label text classifier.
- We propose to enrich the label taxonomy with class-indicative terms, based on which we utilize an embedding-based document-class matching method to improve the pseudo label quality.

- We study two ways of adopting large language models to hierarchical text classification, which can improve the pseudo label quality and solve the data scarcity issue for fine-grained classes.
- Experiments on two datasets show that TELEClass can significantly outperform zero-shot and weakly-supervised hierarchical text classification baselines, while also achieving comparable performance to GPT-4 with drastically less inference cost.[1]

## 2 Problem Definition

The minimally-supervised hierarchical text classification task aims to train a text classifier that can categorize each document into multiple nodes on a label taxonomy by using the name of each node as the only supervision [44]. For example, in Figure 1, the input document is classified as "hair care", "shampoo", and "scalp treatment".

Formally, the task input includes an unlabeled text corpus $\mathcal{D} = \{d_1, \ldots, d_{|\mathcal{D}|}\}$ and a directed acyclic graph (DAG) $\mathcal{T} = (C, \mathcal{R})$ as the label taxonomy. Each $c_i \in C$ represents a target class in the taxonomy, coupled with a unique textual surface name $s_i$. Each edge $\langle c_i, c_j \rangle \in \mathcal{R}$ indicates a hypernymy relation, where class $c_j$ is a subclass of $c_i$. For example, one such edge in Figure 1 is between $s_i =$ "hair care" and $s_j =$ "shampoo". Then, the goal of our task is to train a multi-label text classifier $f(\cdot)$ that can map a document $d$ into a binary encoding of its corresponding classes, $f(d) = [y_1, \ldots, y_{|C|}]$, where $y_i = 1$ represents that $d$ belongs to class $c_i$, otherwise $y_i = 0$.

We assume the label taxonomy to be a DAG instead of a tree, because it aligns better with real applications as one node can have multiple parents with different meanings. It is also more challenging because the classifier needs to assign a document multiple labels in different levels and paths.

## 3 Methodology

In this section, we will introduce TELEClass consisting of the following modules: (1) LLM-enhanced core class annotation, (2) corpus-based taxonomy enrichment, (3) core class refinement with enriched taxonomy, and (4) text classifier training with path-based data augmentation. Figure 2 shows an overview of TELEClass.

## 3.1 LLM-Enhanced Core Class Annotation

Inspired by previous studies, we first tag each document with its "core classes", which are defined as a set of classes that can describe the document most accurately [44]. This process also mimics the process of human performing hierarchical text classification: first select a set of most essential classes for the document and then trace back to their relevant classes to complete the labeling. For example, in Figure 1, by first tagging the document with "shampoo" and "scalp treatment", we can easily find its complete set of classes.

In this work, we propose to enhance the core class annotation process of previous methods with the power of LLMs. To utilize LLMs for core class annotation, we apply a structure-aware candidate core class selection method to reduce the label space for each document. This step is necessary because LLMs can hardly comprehend a large, structured hierarchical label space that cannot be easily represented in a prompt. We first define a similarity score between a class and a document that will be used in candidate

---

[1]Code and datasets are available at: `https://bit.ly/4bWz9h9`

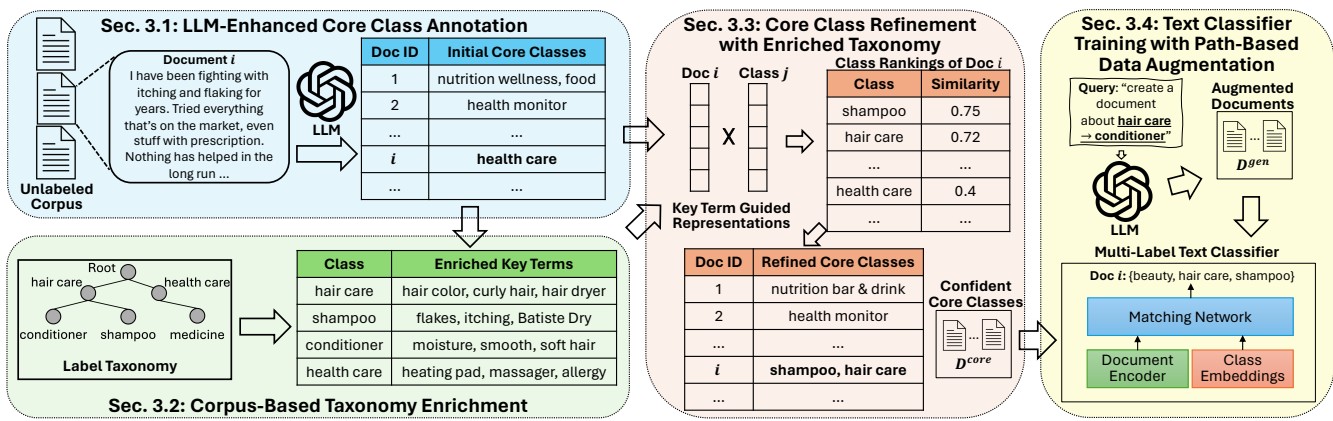

Figure 2: Overview of the TELEClass framework.

selection. To better capture the semantics, we propose to use LLMs to generate a set of class relevant keywords to enrich the raw taxonomy structure and consolidate the meaning of each class. For example, "shampoo" and "conditioner" are two fine-grained classes that are similar to each other. We can effectively separate the two classes by identifying a set of class-specific terms such as "flakes" for "shampoo" and "moisture" for "conditioner". We prompt an LLM to enrich the raw label taxonomy with a set of key terms for each class, denoted as $T_c^{\text{LLM}}$ for the class $c$. To ensure the generated terms can uniquely identify $c$, we ask the LLM to generate terms that are relevant to $c$ and its parent while irrelevant to siblings of $c$. With this enriched set of terms for each class, we define the similarity score between a document $d$ and a class $c$ as the maximum cosine similarity with the key terms:

$$sim(c, d) = \max_{t \in T_c^{\text{LLM}}} \cos(\vec{t}, \vec{d}), \quad (1)$$

where $\vec{\circ}$ denotes a vector representation by a pre-trained semantic encoder (e.g., Sentence Transformer [40]).

This newly defined similarity measure is then used for candidate core class selection. Given a document, we start from the root node at level $l = 0$, select the $l + 3$ most similar children classes to the document at level $l$ using the similarity score defined above, and continue to the next level with only the selected classes. The increasing number of selected nodes accounts for the growing number of classes when going deeper into the taxonomy. Finally, all the classes ever selected in this process will be the candidate core classes for this document, which share the most similarity with the document according to the label hierarchy. [2]

Finally, we instruct an LLM to select the core classes for each document from the selected candidates, which produces an initial set of core classes (denoted as $\mathbb{C}_i^0$) for each document $d_i \in \mathcal{D}$.

## 3.2 Corpus-Based Taxonomy Enrichment

In the previous step, we enrich the raw taxonomy structure with LLM-generated key terms, which are derived from the general knowledge of LLMs but may not accurately reflect the corpus-specific knowledge. Therefore, we propose further enriching the

classes with class-indicative terms mined from the text corpus. By doing this, we can combine the general knowledge of LLMs and corpus-specific knowledge to better enhance the very weak supervision, which is essential for correctly understanding fine-grained classes that are hard to distinguish. Formally, given a class $c \in C$ and its siblings corresponding to one of its parents $c_p$, $Sib(c, c_p) = \{c' \in C | \langle c_p, c' \rangle \in \mathcal{R}\}$, $\mathbf{c}_p \in Par(c)$, we find a set of corpus-based class-indicative terms of $c$ corresponding to $c_p$, denoted as $T(c, c_p) = \{t_1, t_2, \ldots, t_k\}$. Each term in $T(c, c_p)$ can signify the class $c$ and distinguish it from its siblings under $c_p$.

We first collect a set of relevant documents $D_c^0 \subset \mathcal{D}$ for each class $c$, which contains all the documents whose initial core classes contain $c$ or its descendants. Then, inspired by [49, 64], we consider the following three factors for class-indicative term selection and adapt them to the hierarchical setting.

- *Popularity*: a class-indicative term $t$ of a class $c$ should be frequently mentioned by its relevant documents, which is quantified by the log normalization of its document frequency,

$$pop(t, c) = \log(1 + df(t, D_c^0)), \quad (2)$$

where $df(t, D)$ stands for the number of documents in $D$ that mention $t$.

- *Distinctiveness*: a class-indicative term $t$ for a class $c$ should be infrequent in its siblings, which is quantified as the softmax of BM25 relevance function [41] over the set of siblings,

$$dist(t, c, c_p) = \frac{\exp(BM25(t, D_c^0))}{1 + \sum_{c' \in Sib(c, c_p)} \exp(BM25(t, D_{c'}^0))}. \quad (3)$$

- *Semantic similarity*: a class-indicative term $t$ should also be semantically similar to the class name of $c$, which is quantified as the cosine similarity between their embeddings derived from a pre-trained encoder (e.g., BERT [10]), denoted as $sem(c, t)$.

Finally, we define the *affinity* score between a term $t$ and a class $c$ corresponding to parent $p$ to be the geometric mean of the above scores, denoted as $aff(t, c, c_p)$.

To enrich the taxonomy, we first apply a phrase mining tool, AutoPhrase [43], to mine quality single-token and multi-token phrases from the corpus as candidate terms[3]. Then, for each class $c$

---

[2]Refer to Shen et al. [44] for more details on the tree search algorithm.

[3]Our method is flexible with any kinds of phrase mining methods like Gu et al. [19].

and each of its parents $c_p$, we select the top-$k$ terms with the highest affinity scores with $c$ corresponding to $p$, denoted as $T(c, c_p)$. Then, we take the union of these corpus-based terms together with the LLM-generated terms in the previous step to get the final enriched class-indicative terms for class $c$,

$$T_c = \left( \bigcup_{c_p \in Par(c)} T(c, c_p) \right) \bigcup T_c^{\text{LLM}}. \qquad (4)$$

## 3.3 Core Class Refinement with Enriched Taxonomy

With the enriched class-indicative terms for each class, we propose to further utilize them to refine the initial core classes. In this paper, we adopt an embedding-based document-class matching method. Unlike previous methods in flat text classification [51] that use keyword-level embeddings to estimate document and class representations, here, we are able to define class representations directly based on document-level embeddings thanks to the rough class assignments we created in the core class annotation step (c.f. Sec. 3.1).

To obtain document representations, we utilize a pre-trained Sentence Transformer model [40] to encode the entire document, which we denote as $\vec{d}$. Then, for each class $c$, we identify a subset of its assigned documents that explicitly mention at least one of the class-indicative keywords and thus most confidently belong to this class, $D_c = \{d \in D_c^0 | \exists w \in T_c, w \in d\}$. Then, we use the average of their document embeddings as the class representation, $\vec{c} = \frac{1}{|D_c|} \sum_{d \in D_c} \vec{d}$. Finally, we compute the document-class matching score as the cosine similarity between their representations.

Based on the document-class matching scores, we make an observation that the true core classes often have much higher matching scores with the document compared to other classes. Therefore, we use the largest "similarity gap" for each document to identify its core classes. Specifically, for each document $d_i \in \mathcal{D}$, we first get a ranked list of classes according to the matching scores, denoted as $[c_1^i, c_2^i, \ldots, c_{|C|}^i]$, where $\text{diff}^i(j) := \cos(\vec{d}_i, \vec{c}_j^i) - \cos(\vec{d}_i, \vec{c}_{j+1}^i) > 0$ for $j \in \{1, \ldots, |C| - 1\}$. Then, we find the position $m_i$ with the highest similarity difference with its next one in the list. After that, we treat the classes ranked above this position as this document's refined core classes $\mathbb{C}_i$, and the corresponding similarity gap as the confidence estimation $conf_i$.

$$conf_i = \text{diff}^i(m_i), \quad \mathbb{C}_i = \{c_1^i, \ldots, c_{m_i}^i\},$$
$$m_i = \underset{j \in \{1, \ldots, |C|-1\}}{\arg\max} \text{diff}^i(j). \qquad (5)$$

Finally, we select top 75% of documents $d_i$ and their refined core classes with the highest confidence scores $conf_i$, denoted as $\mathcal{D}^{\text{core}}$.

## 3.4 Text Classifier Training with Path-Based Data Augmentation

The final step of TELEClass is to train a hierarchical text classifier using the confident refined core classes. One straightforward way is to directly use the selected core classes as a complete set of pseudo-labeled documents and train a text classifier in a common supervised way. However, such a strategy is ineffective, because

---

**Algorithm 1:** TELEClass

**Input:** A corpus $\mathcal{D}$, a label taxonomy $\mathcal{T}$, a pretrained text encoder $\mathcal{S}$, an LLM $\mathcal{G}$.

**Output:** A text classifier $F$ that can classify each document into a set of classes in $\mathcal{T}$.

1 // LLM-Enhanced Core Class Annotation;
2 **for** $c \in C$ **do**
3    $T_c^{\text{LLM}} \leftarrow$ use $\mathcal{G}$ to enrich $c$ with key terms;
4 **for** $d_i \in \mathcal{D}$ **do**
5    $\mathbb{C}_i^0 \leftarrow$ use $\mathcal{G}$ to select core classes from candidates retrieved using $\mathcal{S}$ and $T_c^{\text{LLM}}$;
6 // Corpus-Based Taxonomy Enrichment;
7 **for** $c \in C$ **do**
8    $D_c^0 \leftarrow$ a set of roughly classified documents;
9    **for** $c_p \in Par(c)$ **do**
10      $T(c, c_p) \leftarrow$ top terms ranked by affinity;
11    $T_c \leftarrow$ aggregate corpus-based and LLM-generated terms Eq. 4;
12 // Core Class Refinement with Enriched Taxonomy;
13 $\vec{d} \leftarrow$ document representation $\mathcal{S}(d)$;
14 **for** $c \in C$ **do**
15    $D_c \leftarrow$ confident documents by matching $T_c$;
16    $\vec{c} \leftarrow$ average document representation in $D_c$;
17 **for** $d_i \in \mathcal{D}$ **do**
18    $\mathbb{C}_i, conf_i \leftarrow$ refined core classes using $\cos(\vec{d}, \vec{c})$ and Eq. 5;
19 $\mathcal{D}^{\text{core}} \leftarrow$ confident refined core classes;
20 // Text Classifier Training with Path-Based Data Augmentation;
21 $\mathcal{D}^{\text{gen}} \leftarrow$ generate $q$ documents for each path using $\mathcal{G}$;
22 $F \leftarrow$ train classifier with $\mathcal{D}^{\text{core}}$ and $\mathcal{D}^{\text{gen}}$;
23 Return $F$;

---

the core classes are not comprehensive enough and cannot cover all the classes in the taxonomy. This is because the hierarchical label space naturally contains fine-grained and long-tail classes, and they are often not guaranteed to be selected as core classes due to their low frequency. Empirically, for the two datasets we use in our experiments, Amazon and DBPedia, the percentages of classes never selected as core classes are 11.6% and 5.4%, respectively. These missing classes will never be used as positive classes in the training process if we only train the classifier with the selected core classes.

Therefore, to overcome this issue, we propose the idea of *path-based document generation by LLMs* to generate a small number of augmented documents (e.g., $q = 5$) for *each* distinct path from a level-1 node to a leaf node in the taxonomy. By adding the generated documents to the pseudo-labeled data, we can ensure that each class of the taxonomy will be a positive class of at least $q$ documents. Because we generate a small constant number of documents for each label path, it also does not affect the distribution of the frequent classes. Moreover, we use a path instead of a single class to guide the LLM generation, because the meaning of lower-level classes is often conditioned on their parents. For example, in Figure 2, a path

**Table 1: Datasets overview.**

| Dataset | # unlabeled train | # test | # labels |
|---------|-------------------|--------|----------|
| Amazon-531 | 29,487 | 19,685 | 531 |
| DBPedia-298 | 196,665 | 49,167 | 298 |

"hair care" → "shampoo" can guide the LLM to generate text about hair shampoo instead of pet shampoo or carpet shampoo that are in different paths. To promote data diversity, we make one LLM query for each path and ask it to generate $q$ diverse documents. We denote the generated documents as $\mathcal{D}^{\text{gen}}$. Appx. A shows the prompts we used.

Now, with two sets of data, the pseudo-labeled documents $\mathcal{D}^{\text{core}}$ and LLM-generated documents $\mathcal{D}^{\text{gen}}$, we are ready to introduce the classifier architecture and the training process.

**Classifier architecture.** We use a simple text matching network similar to [44] as our model architecture, which includes a document encoder initialized with a pre-trained BERT-base model [10] and a log-bilinear matching network. Class representations are initialized by class name embeddings (c.f. Sec. 3.2) and are detached from the encoder model, so only the embeddings will be updated without back-propagation to the backbone model.

**Training process.** For each document with refined core classes, we construct its positive classes as the union of its refined core classes and their ancestors in the label taxonomy, and its negative classes are the ones that are not positive classes or descendants of any core class. This is because the ancestors of confident core classes are also likely to be true labels, and the descendants may not all be negative given that the automatically generated core classes are not optimal. For the LLM-generated documents, we are confident in their labels, so we simply treat all the classes in the corresponding label path as positive classes and all other classes as negative.

We train the text-matching model with the standard binary cross-entropy loss. The loss terms of two sets of data are weighted by their relative size, $\frac{|\mathcal{D}^{\text{core}}|}{|\mathcal{D}^{\text{gen}}|}$. Notice that we do not continue training the classifier with self-training that is commonly used in previous studies [34, 44]. Using self-training may further improve the model performance, which we leave for future exploration. Algorithm 1 summarizes TELEClass.

## 4 Experiments

### 4.1 Experiment Setup

*4.1.1 Datasets.* We use two public datasets in different domains for evaluation. Table 1 shows the data statistics.

- **Amazon-531** [31] consists of Amazon product reviews and a three-layer label taxonomy of product types.
- **DBPedia-298** [28] consists of Wikipedia articles with a three-layer label taxonomy of its categories.

*4.1.2 Compared Methods.* We compare the following methods on the weakly-supervised hierarchical text classification task.

- **Hier-0Shot-TC** [55] is a zero-shot approach, which utilizes a pretrained textual entailment model to iterative find the most similar class at each level for a document.

- **GPT-3.5-turbo** is a zero-shot approach that queries GPT-3.5-turbo by directly providing all classes in the prompt.
- **Hier-doc2vec** [27] is a weakly-supervised approach, which first trains document and class representations in a same embedding space, and then iteratively selects the most similar class at each level.
- **WeSHClass** [34] is a weakly-supervised approach using a set of keywords for each class. It first generates pseudo documents to pretrain text classifiers and then performs self-training.
- **TaxoClass** [44] is a weakly-supervised approach that only uses the class name of each class. It first uses a textual entailment model with a top-down search and corpus-level comparison to select core classes, which are then used as pseudo training data. We include both its full model, TaxoClass, and its variation TaxoClass-NoST that does not apply self-training on the trained classifier, which is the same as TELEClass.
- **TELEClass** is our newly proposed weakly-supervised approach that only uses the class name for each class.
- **Fully-Supervised** is a fully-supervised baseline that uses the entire labeled training data to train the text matching network used in TELEClass.

*4.1.3 Evaluation Metrics.* Following previous studies [44], we utilize the following evaluation metrics:

- **Example-F1** [47], which is also called micro-Dice coefficient, evaluates the multi-label classification results without ranking,

$$\text{Example-F1} = \frac{1}{|\mathcal{D}|} \sum_{d_i \in \mathcal{D}} \frac{2 \cdot |\mathbb{C}_i^{\text{true}} \cap \mathbb{C}_i^{\text{pred}}|}{|\mathbb{C}_i^{\text{true}}| + |\mathbb{C}_i^{\text{pred}}|}, \quad (6)$$

where $\mathbb{C}_i^{\text{true}}$ and $\mathbb{C}_i^{\text{pred}}$ denote the set of true labels and the set of predicted labels for document $d_i \in \mathcal{D}$, respectively.

- **Precision at k**, or **P@k**, is a ranking-based metric that evaluates the precision of top-$k$ predicted classes,

$$\text{P@k} = \frac{1}{k} \sum_{d_i \in \mathcal{D}} \frac{|\mathbb{C}_i^{\text{true}} \cap \mathbb{C}_{i,k}^{\text{pred}}|}{\min(k, |\mathbb{C}_i^{\text{true}}|)}, \quad (7)$$

where $\mathbb{C}_{i,k}^{\text{pred}} \subset \mathbb{C}_i^{\text{pred}}$ denotes the top-$k$ predicted labels if the evaluated method can generate rankings of classes.

- **Mean Reciprocal Rank**, or **MRR**, is another ranking-based metric, which evaluates the multi-label predictions based on the inverse of true labels' ranks within predicted classes,

$$\text{MRR} = \frac{1}{|\mathcal{D}|} \sum_{d_i \in \mathcal{D}} \frac{1}{|\mathbb{C}_i^{\text{true}}|} \sum_{c_j \in \mathbb{C}_i^{\text{true}}} \frac{1}{\min\{k | c_j \in \mathbb{C}_{i,k}^{\text{pred}}\}}. \quad (8)$$

*4.1.4 Implementation Details.* We use Sentence Transformer [40] `all-mpnet-base-v2` as the text encoder for the similarity measure in Section 3.1 and Section 3.3. We query `GPT-3.5-turbo-0125` for LLM-based taxonomy enrichment, core class annotation, and path-based generation. For corpus-based taxonomy enrichment, we get the term and class name embeddings using a pre-trained `BERT-base-uncased` [10], we select top $k = 20$ enriched terms for each class (c.f. Section 3.2). We generate $q = 5$ documents with path-based generation for each class. The document encoder in the final classifier is initialized with `BERT-base-uncased` for a fair comparison with the baselines. We train the classifier using AdamW

**Table 2: Experiment results on Amazon-531 and DBPedia-298 datasets, evaluated by Example-F1, P@k, and MRR. The best score among zero-shot and weakly-supervised methods is boldfaced. "†" indicates the numbers for these baselines are directly from previous paper [44]. "—" means the method cannot generate a ranking of predictions and thus MRR cannot be calculated.**

| Supervision Type | Methods | Amazon-531 | | | | DBPedia-298 | | | |
|---|---|---|---|---|---|---|---|---|---|
| | | Example-F1 | P@1 | P@3 | MRR | Example-F1 | P@1 | P@3 | MRR |
| Zero-Shot | Hier-0Shot-TC† | 0.4742 | 0.7144 | 0.4610 | — | 0.6765 | 0.7871 | 0.6765 | — |
| | ChatGPT | 0.5164 | 0.6807 | 0.4752 | — | 0.4816 | 0.5328 | 0.4547 | — |
| Weakly-Supervised | Hier-doc2vec† | 0.3157 | 0.5805 | 0.3115 | — | 0.1443 | 0.2635 | 0.1443 | — |
| | WeSHClass† | 0.2458 | 0.5773 | 0.2517 | — | 0.3047 | 0.5359 | 0.3048 | — |
| | TaxoClass-NoST† | 0.5431 | 0.7918 | 0.5414 | 0.5911 | 0.7712 | 0.8621 | 0.7712 | 0.8221 |
| | TaxoClass† | 0.5934 | 0.8120 | 0.5894 | 0.6332 | 0.8156 | 0.8942 | 0.8156 | 0.8762 |
| | TELEClass | **0.6483** | **0.8505** | **0.6421** | **0.6865** | **0.8633** | **0.9351** | **0.8633** | **0.8864** |
| Fully-Supervised | | 0.8843 | 0.9524 | 0.8758 | 0.9085 | 0.9786 | 0.9945 | 0.9786 | 0.9826 |

optimizer with a learning rate 5e-5, and the batch size is 64. The experiments are run on one NVIDIA RTX A6000 GPU.

## 4.2 Experimental Results

Table 2 shows the evaluation results of all the compared methods. We make the following observations. (1) Overall, TELEClass achieves significantly better performance than other strong zero-shot and weakly-supervised baselines, which demonstrates the effectiveness of TELEClass on the hierarchical text classification task without any human supervision. (2) By comparing with other weakly-supervised methods, we find that TELEClass significantly outperforms TaxoClass-ST, the strongest baseline that, like TELEClass, does not use self-training. Given that TELEClass uses an even simpler classifier model than TaxoClass-ST, its superior performance shows the substantially better pseudo training data obtained by combining unlabeled corpus and LLMs. (3) Although LLMs (e.g., ChatGPT) show power in many tasks, naïvely prompting it in the hierarchical text classification task yields significantly inferior performance compared to strong weakly-supervised text classifiers. This proves the necessity of incorporating corpus-based knowledge to improve label taxonomy understanding for the hierarchical setting. We conduct a more detailed comparison with LLM prompting for hierarchical text classification in Section 4.4.

We also study the temporal complexity of TELEClass. We observe that, on Amazon-531, both TELEClass and the strongest baseline TaxoClass take around 5 to 5.5 hours. The reason why TELEClass does not increase the overall temporal complexity is that TaxoClass needs to run the textual entailment model on each pair of document and candidate class. On the other hand, the taxonomy enrichment step of TELEClass makes it possible to simplify this process with embedding similarity calculation which saves a lot of time, while the saved time is budgeted for LLM prompting.

## 4.3 Ablation Studies

We conduct ablation studies to better understand how each component of TELEClass contributes to final performance. Table 3 shows the results of the following ablations:

- **Gen-Only** only uses the augmented documents by path-based LLM generation to train the final classifier.
- **TELEClass-NoLLMEnrich** excludes the LLM-based taxonomy enrichment component.
- **TELEClass-NoCorpusEnrich** excludes the corpus-based taxonomy enrichment component.
- **TELEClass-NoGen** excludes the augmented documents by path-based LLM generation.

We find that the full model TELEClass achieves the overall best performance among the compared methods, showing the effectiveness of each of its components. First, both the LLM-based and corpus-based enrichment modules bring improvement to the performance. Interestingly, we find that they make different levels of contribution on the two datasets: LLM-based enrichment brings more improvement on Amazon-531 while corpus-based enrichment contributes more on DBPedia-298. We suspect the reasons are as follows. The classes in Amazon-531 are commonly seen product types that LLM can understand and enrich in a reliable manner. However, DBPedia-298 contains classes that are more subtle to distinguish, which can also be shown by the lower performance of zero-shot LLM prompting on DBPedia compared to Amazon-531 (c.f. ChatGPT in Table 2). Therefore, corpus-based enrichment can consolidate the meaning of each class based on corpus-specific knowledge to facilitate better classification. We also find that path-based LLM generation consistently improves the model performance while requiring only a few hundred queries to LLMs. Even Gen-Only achieves comparable performance to the strong baseline TaxoClass-NoST, demonstrating the effectiveness of this augmentation step.

## 4.4 Comparison with Zero-Shot LLM Prompting

In this section, we further compare TELEClass with zero-shot LLM prompting. Because it is not straightforward to get ranked predictions by LLMs, we only report Example-F1 and P@k as performance evaluation. Additionally, we report the estimated cost and time for each method on the entire test set. The inference time is reported in minutes, and please be aware that this is just a rough estimation as the actual running time is also dependent on the server condition.

We include the following settings to compare with TELEClass:

**Table 3: Performance of TELEClass and its ablations on Amazon-531 and DBPedia-298 datasets. The best score is boldfaced.**

| Methods | Amazon-531 | | | | DBPedia-298 | | | |
|---|---|---|---|---|---|---|---|---|
| | Example-F1 | P@1 | P@3 | MRR | Example-F1 | P@1 | P@3 | MRR |
| Gen-Only | 0.5151 | 0.7477 | 0.5096 | 0.5357 | 0.7930 | **0.9421** | 0.7930 | 0.8209 |
| TELEClass-NoLLMEnrich | 0.5520 | 0.7370 | 0.5463 | 0.5900 | 0.8319 | 0.9108 | 0.8319 | 0.8563 |
| TELEClass-NoCorpusEnrich | 0.6143 | 0.8358 | 0.6082 | 0.6522 | 0.8185 | 0.8916 | 0.8185 | 0.8463 |
| TELEClass-NoGen | 0.6449 | 0.8348 | 0.6387 | 0.6792 | 0.8494 | 0.9187 | 0.8494 | 0.8730 |
| TELEClass | **0.6483** | **0.8505** | **0.6421** | **0.6865** | **0.8633** | 0.9351 | **0.8633** | **0.8864** |

**Table 4: Performance comparison of TELEClass and zero-shot LLM prompting. We only report Example-F1 and P@k, because it is not straightforward to get ranking of classes predicted by LLMs for MRR calculation. We also report estimated costs in US dollars and running time in minutes for each method on the entire test set. "‡" indicates that we report the performance based on an estimation from a 1,000-document subset of test data.**

| Methods | Amazon-531 | | | | | DBPedia-298 | | | | |
|---|---|---|---|---|---|---|---|---|---|---|
| | Example-F1 | P@1 | P@3 | Est. Cost | Est. Time | Example-F1 | P@1 | P@3 | Est. Cost | Est. Time |
| GPT-3.5-turbo | 0.5164 | 0.6807 | 0.4752 | $60 | 240 mins | 0.4816 | 0.5328 | 0.4547 | $80 | 400 mins |
| GPT-3.5-turbo (level) | 0.6621 | 0.8574 | 0.6444 | $20 | 800 mins | 0.6649 | 0.8301 | 0.6488 | $60 | 1,000 mins |
| GPT-4‡ | 0.6994 | 0.8220 | 0.6890 | $800 | 400 mins | 0.6054 | 0.6520 | 0.5920 | $2,500 | 1,000 mins |
| TELEClass | 0.6483 | 0.8505 | 0.6421 | <$1 | 3 mins | 0.8633 | 0.9351 | 0.8633 | <$1 | 7 mins |

- **GPT-3.5-turbo**: We include all the classes in the prompt and ask GPT-3.5-turbo model to provide 3 most appropriate classes for a given document.
- **GPT-3.5-turbo (level)**: We perform level-by-level prompting using GPT-3.5-turbo. Starting from the root node, we ask the model to return one most appropriate class for a given document, and we iteratively prompt the model with the children of the selected node at each level. This method can only generate a path in the taxonomy, but in the actual multi-label hierarchical classification setting, the true labels may not sit in the same path.
- **GPT-4**: Similar to the first one, we include all the classes in the prompt and ask GPT-4 to provide 3 most appropriate classes for a given document. Given the limited budget, we only test it on randomly sampled 1,000 documents and estimate the cost on the entire test set.

Table 4 shows the experiment results. We find that TELEClass consistently outperforms all compared methods on DBPedia, while on Amazon, TELEClass underperforms GPT-3.5-turbo (level) and GPT-4 but still being comparable. As for the cost, once trained, TELEClass does not require additional cost on inference and also has substantially shorter inference time. Prompting LLMs takes longer time and can be prohibitively expensive (e.g., using GPT-4), and the cost will scale up with increasing size of test data. Also, we find that GPT-3.5-turbo (level) consistently outperforms the naïve version, demonstrating the necessity of taxonomy structure. It saves the cost because of the much shorter prompts, but takes longer time due to more queries made per document.

## 4.5 Case studies

To better understand the TELEClass framework, we show some intermediate results of two documents in Table 5, including the

core classes selected by (1) the original TaxoClass [44] method, (2) TELEClass's initial core classes selected by LLM, and (3) refined core classes of TELEClass. Besides, we also include the true labels of the documents and the taxonomy enrichment results of the corresponding core class in the table. Overall, we can see that TELEClass's refined core class is the most accurate. For example, for the first Wikipedia article about a library, TaxoClass selects "village" as the core class, while TELEClass's initial core class finds a closer one "building" thanks to the power of LLMs. Then, with the enriched classes-indicative features as guidance, TELEClass's refined core class correctly identifies the optimal core class, which is "library". In the other example, TELEClass also pinpoints the most accurate core class "bathroom aids safety" while other methods can only find more general or partially relevant classes.

Besides, although TELEClass outperforms the zero-shot LLM prompting in most cases, there are cases showing the contrary and here is one example we found from Amazon-531. The product review is about "glycolic treatment pads" for which GPT correctly predicts its labels as "beauty" and "skin care", while TELEClass predicts it as "health care". We suspect that the word "treatment" in the review leads to the error because of the bias of term-based pseudo-labeling. It is a known issue of keyword-based methods and some solutions are proposed for the weakly-supervised flat text classification [12, 62]. We hope our study can motivate more research to solve this issue in the hierarchical setting.

## 5 Related Work

### 5.1 Weakly-Supervised Text Classification

Weakly supervised text classification trains a classifier with limited guidance, aiming to reduce human efforts while maintaining high proficiency. Various sources of weak supervision have been

**Table 5: Intermediate results on two documents, including selected core classes by different methods, true labels, and the corresponding taxonomy enrichment results. The optimal core class in the true labels is marked with ©.**

| Dataset | Document | Core Classes by... | True Labels | Corr. Enrichment |
|---------|----------|--------------------|-------------|------------------|
| DBPedia | The Lindenhurst Memorial Library (LML) is located in Lindenhurst, New York, and is one of the fifty six libraries that are part of the Suffolk Cooperative Library System ... | **TaxoClass**: village
**TELEClass initial**: building
**TELEClass refined**: library | library©, agent, educational institution | **Class**: library
**Top Enrichment**: national library, central library, collection, volumes... |
| Amazon | Since mom (89 yrs young) isn't steady on her feet, we have placed these grab bars around the room. It gives her the stability and security she needs. | **TaxoClass**: personal care, health personal care, safety
**TELEClass initial**: daily living aids, medical supplies equipment, safety
**TELEClass refined**: bathroom aids safety | health personal care, medical supplies equipment, bathroom aids safety© | **Class**: bathroom aids safety
**Top Enrichment**: seat, toilet, shower, safety, handles... |

explored, including distant supervision [8, 17, 46] like knowledge bases, keywords [1, 33, 35, 49, 51, 59], and heuristic rules [3, 39, 45]. Later, extremely weakly-supervised methods are proposed to solely rely on class names to generate pseudo labels and train classifiers. LOTClass [35] utilizes MLM-based PLM as a knowledge base for extracting class-indicative keywords. X-Class [51] extracts keywords for creating static class representations through clustering. PIEClass [62] employs PLMs' zero-shot prompting to obtain pseudo labels with noise-robust iterative ensemble training. MEGClass [26] acquires contextualized sentence representations to capture topical information at the document level. WOTClass [50] ranks class-indicative keywords from generated classes and extracts classes with overlapping keywords.

### 5.2 Hierarchical Text Classification

A hierarchy provides a systematic top-down structure with inherent semantic relations that can assist in text classification. Typical hierarchical text classification can be categorized into two groups: local approaches and global approaches. Local approaches train multiple classifiers for each node or local structures [4, 30, 53]. Global approaches, learn hierarchy structure into a single classifier through recursive regularization [18], a graph neural network (GNN)-based encoder [24, 37, 44, 65], or a joint document label embedding space [9]. Recent studies also show that LLMs cannot comprehend the complex hierarchical structure [6, 16, 57].

Weak supervision is also studied for hierarchical text classification. WeSHClass [34] uses a few keywords or example documents per class and pretrains classifiers with pseudo documents followed by self-training. TaxoClass [44] follows the same setting as ours which uses the sole class name of each class as the only supervision. It identifies core classes for each document using a textual entailment model, which is then used to train a multi-label classifier. Additionally, MATCH [63] and HiMeCat [61] study how to integrate associated metadata into the label hierarchy for document categorization with weak supervision.

### 5.3 LLMs as Generators and Annotators

Large language models (LLMs) have demonstrated impressive performance in many downstream tasks and are explored to help low-resource settings by synthesizing data as generators or annotators [11]. For data generation, few-shot examples [52] or class-conditioned prompts [32] are explored for LLM generation and the generated data can be used as pseudo training data to further fine-tune a small model as the final classifier [54]. Recently, Yu et al. [58] proposed an attribute-aware topical text classification method that incorporates ChatGPT to generate topic-dependent attributes and topic-independent attributes to reduce topic ambiguity and increase topic diversity for generation. For data annotation, previous works utilize LLMs for unsupervised annotation [15], Chain-of-Thought annotation with explanation generation [22], and active annotation [60].

## 6 Conclusion and Future Work

In this paper, we propose a new method, TELEClass, for the minimally-supervised hierarchical text classification task with two major contributions. First, we enrich the input label taxonomy with LLM-generated and corpus-based class-indicative terms for each class, which can serve as additional features to understand the classes and facilitate classification. Second, we explore the utilization of LLMs in the hierarchical text classification in two directions: data annotation and data creation. On two public datasets, TELEClass can outperform existing baselines substantially, and we further demonstrate its effectiveness through ablation studies. We also conduct a comparative analysis of performance and cost for zero-shot LLM prompting for the hierarchical text classification task.

For future works, first, we plan to generalize TELEClass's idea of combining LLMs with data-specific knowledge into other low-resource text mining tasks with hierarchical label spaces, such as fine-grained entity typing. Second, in this paper, we mainly focus on acquiring high-quality pseudo labeled data while only utilizing the simplest classifier model and objective. It is worth studying how the proposed method can be further improved with more advanced network structure and noise-robust training objectives. Lastly, we also plan to explore how to extend TELEClass into harder settings like when existing LLMs do not have the knowledge for the initial annotation (e.g., a private domain), a lower-resource scenario where the availability of the unlabeled corpus is limited, or a more complicated label space like an extremely large hierarchical label space with millions of classes.

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

# A Prompts for LLM

- **LLM-based enrichment for Amazon-531**

> **Instruction:** [Target Class] is a product class in Amazon and is the subclass of [Parent Class]. Please generate 10 additional key terms about the [Target Class] that are relevant to [Target Class] but irrelevant to [Sibling Classes]. Please split the additional key terms using commas.

- **LLM-based enrichment for DBPedia-298**

> **Instruction:** [Target Class] is an article category of Wikipedia articles and is the subclass of [Parent Class]. Please generate 10 additional key terms about the [Target Class] that are relevant to [Target Class] but irrelevant to [Sibling Classes]. Please split the additional key terms using commas.

- **Core class annotation for Amazon-531**

> **Instruction:** You will be provided with an Amazon product review, and please select its product types from the following categories: [Candidate Classes]. Just give the category names as shown in the provided list.
> **Query:** [Document]

- **Core class annotation for DBPedia-298**

> **Instruction:** You will be provided with a Wikipedia article describing an entity at the beginning, and please select its types from the following categories: [Candidate Classes]. Just give the category names as shown in the provided list.
> **Query:** [Document]

- **Path-based generation for Amazon-531**

> **Instruction:** Suppose you are an Amazon Reviewer, please generate 5 various and reliable passages following the requirements below:
> 1. Must generate reviews following the themes of the taxonomy path: [Path].
> 2. Must be in length about 100 words.
> 3. The writing style and format of the text should be a product review.
> 4. Should keep the generated text to be diverse, specific, and consistent with the given taxonomy path. You should focus on [The Leaf Node on the Path].

- **Path-based generation for DBPedia-298**

> **Instruction:** Suppose you are a Wikipedia Contributor, please generate 5 various and reliable passages following the requirements below:
> 1. Must generate reviews following the themes of the taxonomy path: [Path].
> 2. Must be in length about 100 words.
> 3. The writing style and format of the text should be a Wikipedia page.
> 4. Should keep the generated text to be diverse, specific, and consistent with the given taxonomy path. You should focus on [The Leaf Node on the Path].

