# OpenReview forum: "TELEClass: Taxonomy Enrichment and LLM-Enhanced Hierarchical Text Classification with Minimal Supervision"
_ACM.org/TheWebConf/2025/Conference — WWW 2025 Poster_

### Official Review · Reviewer_vfiH · 2024-11-24

**Novelty:** 3
**Technical Quality:** 4

**Review:**

This paper introduces TELEClass, a new method for minimally supervised hierarchical text classification. The key challenges addressed are understanding the large structured label space and distinguishing the semantics of similar classes, while only using the class names as supervision. TELEClass combines the general knowledge of large language models (LLMs) and task-specific features mined from an unlabeled corpus to tackle these challenges. It automatically enriches the raw taxonomy with class-indicative features, utilizes novel LLM-based data annotation and generation methods tailored for the hierarchical setting, and trains a multi-label text classifier on the pseudo-labeled data. Experiments show that TELEClass can significantly outperform previous baselines while also achieving comparable performance to zero-shot prompting of LLMs with drastically less inference cost.

Pros:

1.It proposes a new method called TELEClass for minimally supervised hierarchical text classification, which addresses the challenges of understanding the large structured label space and distinguishing the semantics of similar classes, while only using the class names as supervision.

2.Experiments show that TELEClass can significantly outperform previous baselines while also achieving comparable performance to zero-shot prompting of LLMs with drastically less inference cost.

3.The paper is well-organized and written in a rigorous style, showcasing excellent presentation skills.

Questions & Cons:

1.In the first module, I am concerned about how to ensure the key term generated by LLM is not a target class in C.

2.Try using more LLM models, such as LLama 3 in enrichment, and evaluate the performance during the experiments, which might yield better results compared to GPT 3.5.

**Questions:**

1.In the first module, I am concerned about how to ensure the key term generated by LLM is not a target class in C.

2.Try using more LLM models, such as LLama 3 in enrichment, and evaluate the performance during the experiments, which might yield better results compared to GPT 3.5.

**Reviewer Confidence:**

2: The reviewer is willing to defend the evaluation, but it is likely that the reviewer did not understand parts of the paper

**Scope:**

3: The work is somewhat relevant to the Web and to the track, and is of narrow interest to a sub-community

---

### Official Review · Reviewer_YanF · 2024-12-01

**Novelty:** 2
**Technical Quality:** 3

**Review:**

This paper proposes TELEClass, a weakly-supervised hierarchical text classification method that addresses the hierarchical classification problem with minimal supervision (only using class names). TELEClass combines the general knowledge of large language models (LLMs) and task-specific features mined from an unlabeled corpus to automatically enrich the raw taxonomy with class-indicative features, improving label space understanding. The method leverages LLM-based data annotation and generation techniques tailored for the hierarchical setting. Experiments show that TELEClass outperforms previous strong baselines while achieving comparable performance to LLM zero-shot prompting.

- pros

(1) The experiments in this paper are quite thorough, comparing the proposed method with various strong baseline models. Additionally, it also includes a performance comparison with closed-source models.

(2) The paper provides a clear explanation of related work.

- cons

(1) The writing of this paper needs improvement, as it is somewhat difficult to read and understand in the methodology section.

(2) From Figure 1 and Figure 2, I can't clearly discern the specific approach being used, as many categories and labels appear abruptly. The authors need to rethink how to present them more clearly and make the methodology more intuitive.

(3) According to my understanding, the paper utilizes large language models (LLMs) to generate categories and labels, then ranks and filters them through some similarity metrics. The remaining category label paths are used to generate document. The entire process relies on heuristic rules and LLMs to construct the data. It seems that the technical contribution is somewhat weak.

**Questions:**

(1) The authors could explain the entire pipeline of the method in a few sentences.

(2) What is the relationship between Initial Core Classes, Enriched Key Terms, and Refined Core Classes?

**Reviewer Confidence:**

3: The reviewer is confident but not certain that the evaluation is correct

**Scope:**

4: The work is relevant to the Web and to the track, and is of broad interest to the community

---

### Official Review · Reviewer_WMJy · 2024-12-02

**Novelty:** 5
**Technical Quality:** 5

**Review:**

This paper proposes the TELEClass method for taxonomy enrichment and LLM-enhanced hierarchical text classification with minimal supervision. It combines the knowledge of LLM and the features mined from the unlabeled corpus through a series of steps such as LLM-enhanced core class annotation, corpus-based taxonomy enrichment, core class refinement, and text classifier training with path-based data augmentation. The apporach solves the problems faced by traditional methods in hierarchical text classification, such as the difficulty in understanding the large-scale structured label space, the high cost of data annotation, and the poor performance of LLM in the hierarchical setting. Experiments on two datasets demonstrate its superiority over existing methods, including zero-shot prompting of LLMs.

Pros:
1. The research addresses the key issue of weak supervision in hierarchical text classification. The proposed TELEClass method provides new ideas and effective solutions in this field, which plays a crucial role in promoting the development of text classification technology, especially how to utilize LLM to extract task-specific knowledge from unlabeled corpus.
2. The idea of first accouting for core classes and then refining them can be extended to other scenarios when a high-quality training dataset needs to be prepared with a large number of (hierarchical) classes.
3. The experimental design is comprehensive. The evaluation is conducted on two public datasets in different domains, and a variety of related methods, including zero-shot and weakly-supervised methods, are compared. The rich evaluation metrics can fully prove the effectiveness and superiority of the TELEClass method.
4. The case study with intermediate results is beneficial to understand the execution flow and direct effects of the approach.
5. The structure and presentation of the paper is very good. The idea is easy to understand with almost no typos.

Cons:
1. In section 3.4, there is an overreliance on the background knowledge of TaxoClass. For example, the description of positive and negative class definitions and the choice of not using self-training lacks sufficient explanation, making it difficult for readers unfamiliar with the related work to understand the last step as well as some ideas in previous steps. Also, the formula for the final loss function is not presented, which makes it challenging for readers unfamiliar with the related research to grasp the underlying principles, especially for the multi-label text classification task.
2. The core steps are composed of many human-crafted rules, and there is a lack of in-depth analysis and discussion on the impact of some key designs and hyperparameters in the method, such as l+3 most similar classes in Section 3.1, the geometric mean and the selection of top-k terms in Section 3.2, 75% in Section 3.3, and q=5 in Section 3.4). That affects the interpretability and extensibility of the method.
3. The motivating figure (Figure 1) is better to be substituted as a DAG instead of a tree, to be consistent with the problem formulation.
4. The definition of multi-label text classification in line 204 is not clear. Does the classifier need to assign a label in each level?
5. Since a core innovation of the paper is the utilization of LLM to understand unlabeled corpus and enrich the semantics of classes, some latest papers on this topic in weakly-supervised text classification should be discussed, such as NPPrompt, PESCO, WDDC and RulePrompt. They also employ LLMs to generate pseudo labels and/or mine rules to accurately characterize each class, which further serve as new prompts of LLMs.

**Questions:**

1. In practical applications, how to ensure the accuracy and consistency of the class-related keywords generated by LLM in Step 2, especially for text classification tasks in some professional or emerging fields?
2. How to determine the hyperparameters mentioned in Cons 2 above. For example, for the path-based data augmentation, what is the consideration for setting the number of augmented documents as  q = 5, and is it universal for datasets of different scales or domains?

**Reviewer Confidence:**

4: The reviewer is certain that the evaluation is correct and very familiar with the relevant literature

**Scope:**

4: The work is relevant to the Web and to the track, and is of broad interest to the community

---

### Official Review · Reviewer_5tYP · 2024-12-03

**Novelty:** 3
**Technical Quality:** 3

**Review:**

This paper proposes a method for hierarchical text classification. The idea is simple, and the paper is easy to read.

There are many details that are missed, but it is necessary for readers to understand the details of the experiment.

First, the strong baseline is from 2021's paper. This is problematic for a submission for the 2025 top conference. Readers cannot learn whether the performance is significantly different between the proposed methods and recent well-performing methods.

Second, the comparison is unfair. The proposed model utilizes LLM in the loop, but the baseline methods are the methods before we have LLMs. That makes the experimental results are not convincing.

Third, Only "GPT-3.5" was used for major experiments. To understand the robustness of the proposed method, using more LLM with the proposed approach is necessary. It could help readers know whether the claims only happen when using GPT-3.5 or if it is a general finding regardless of the LLM used.

Fourth, Only exploring zero-shot LLM is not enough to reflect the development of recent LLM discussions. The concept of "Supervision" was used in the proposed method. It should also be adopted when using LLMs, e.g., few-shot.

Fifth, the results of GPT-4 in Table 4 are not comparable with others because the setting is different.

Overall, the idea is similar to other studies that use LLM for data augmentation, and this paper pays attention to the applications of hierarchical classification. However, the experiment is not sufficient to support that the proposed method could outperform the state-of-the-art approaches (which is the focus of this paper).

**Questions:**

There are many recent papers, especially those using LLMs in the related work section. Why they cannot be adopted for comparison? How do you use the findings of previous studies to design your approach and prompts?

**Reviewer Confidence:**

3: The reviewer is confident but not certain that the evaluation is correct

**Scope:**

3: The work is somewhat relevant to the Web and to the track, and is of narrow interest to a sub-community

---

### Official Review · Reviewer_Q7GY · 2024-12-03

**Novelty:** 6
**Technical Quality:** 6

**Review:**

Unlike previous methods like TaxoClass, which rely solely on class names, TELEClass enriches the label taxonomy with class-indicative terms. These terms are derived from two sources: LLM generation and automated extraction from the corpus. This enrichment process combines general knowledge from LLMs with specific insights from the text corpus, leading to a more nuanced understanding of the label space and improved pseudo-label quality for classifier training. This paper presents several novel contributions to the field of minimally supervised hierarchical text classification, particularly in its innovative use of taxonomy enrichment, LLM integration, and embedding-based core class refinement techniques. These novelties allow TELEClass to achieve significant performance improvements over existing baselines, offering a promising solution for hierarchical text classification with minimal human supervision.

**Questions:**

1. The paper claims that TELEClass requires only the class names as supervision to train a multi-label text classifier. However, the paper mentions utilising external knowledge sources like LLMs and pre-trained semantic encoder models. To what extent does TELEClass rely on these external resources? Could it truly be considered "minimally supervised" if it depends on the availability and pre-training of these extensive models?

2. Could a more sophisticated prompting strategy, possibly incorporating elements of TELEClass's taxonomy enrichment, further enhance the performance of zero-shot LLM methods?

**Reviewer Confidence:**

3: The reviewer is confident but not certain that the evaluation is correct

**Scope:**

3: The work is somewhat relevant to the Web and to the track, and is of narrow interest to a sub-community